# Ultrasonic Sensor Modeling with Support Vector Regression

**DOI:** 10.3390/s25030678

**Published:** 2025-01-23

**Authors:** Duy Ngoc Dang, Tri Minh Do, Rui Alexandre de Matos Araújo, Khang Hoang Vinh Nguyen, Can Duy Le

**Affiliations:** 1Electrical and Computer Engineering, Vietnamese-German University, Ben Cat 75000, Binh Duong, Vietnam; 14445@student.vgu.edu.vn (D.N.D.); 15752@student.vgu.edu.vn (T.M.D.); 2Institute of Systems and Robotics (ISR-UC), Department of Electrical and Computer Engineering (DEEC-UC), University of Coimbra, Pólo II, 3030-290 Coimbra, Portugal; rui@isr.uc.pt; 3Mechatronics and Sensor Systems Technology, Vietnamese-German University, Ben Cat 75000, Binh Duong, Vietnam; can@vgu.edu.vn

**Keywords:** support vector regression, digital twin, virtual sensor, kernel selection, hyperparameter optimization

## Abstract

This study proposes a novel approach for predicting the output behaviors of the Pepperl+Fuchs 3RG6232-3JS00-PF ultrasonic sensor. The sensor, integrated into the Festo MPS-PA Didactic System, serves to monitor the water level in a tank, facilitating water extraction to bottles delivered via a conveyor belt. This modeling approach represents the initial phase in the creation of a digital twin of the physical sensor, providing the capability for users to observe the sensor’s response and forecast its life cycle for maintenance objectives. This study utilizes the Festo MPS-PA Compact Didactic System and support vector regression (SVR) for data acquisition (DAQ), preprocessing, and model training with hyperparameter optimization. The objective of this modeling approach is to establish a digital framework for transition towards Industry 4.0. It holds the potential for creating a digital counterpart of the entire MPS-PA System when combining the proposed sensor modeling technique with computer-assisted design (CAD) software such as Siemens NX in the future. This would enable users to oversee the entire process in a three-dimensional visualization engine, such as Tecnomatix Plant Simulation. This research significantly contributes to the comprehension and application of digital twins in the realm of mechatronics and sensor systems technology. It also underscores the importance of digital twins in enhancing the efficiency and predictability of sensor systems. The method used in this paper involves predicting the rate of change (RoC) of the water level and then integrating this rate to estimate the actual water level, providing a robust approach for sensor data modeling and digital twin creation. The result shows a promising 6.99% error percentage.

## 1. Introduction

Industrial automation thrives on precise and reliable measurements. Ultrasonic sensors have emerged as a cornerstone technology for diverse tasks, from meticulously gauging liquid levels in tanks to measuring material thickness and determining object proximity with high accuracy. Their operation revolves around emitting high-frequency sound waves and measuring the time it takes for echoes to return. This allows for precise distance calculations and characterization of material properties. Some notable achievements include the development of advanced ultrasonic imaging techniques and the integration of ultrasonic sensors with automated control systems to enhance process efficiency.

Beyond these sensor advancements, machine learning is playing a crucial role in enhancing the performance of industrial measurements using ultrasonic sensors. By analyzing the complex data generated by these sensors, machine learning can identify subtle patterns and trends that would be difficult to discern with traditional methods [1]. Techniques such as neural networks, support vector machine (SVM), and decision trees have been employed to optimize measurement accuracy and detect anomalies. Furthermore, machine learning models can be trained to predict equipment wear and tear based on sensor data, allowing for scheduled maintenance and preventing unplanned downtime.

This research delves into the exciting realm of data-driven approaches, specifically by leveraging machine learning within a digital twin framework. Digital twins are virtual representations of physical assets and systems that integrate real-time data from sensors and other sources. Traditionally, these systems relied on physical-based models to describe the behaviors of sensors and actuators. These models rely on engineering principles and require significant upfront knowledge of the system dynamics. However, a recent shift is happening in the state-of-the-art of digital twins. There is growing interest in data-driven approaches that leverage real-world data to describe sensor and actuator behavior [2]. This study proposes exploring data-driven approaches within a digital twin framework, aiming to investigate on improving the overall performance and reliability of digital twins across various industrial applications.

The digital twin was constructed within the TIA Portal (V17, Siemens AG) software [3] using the Festo MPS PA Didactic System. TIA Portal’s WinCC Runtime Advanced serves as the development environment for the PLC program controlling the station, including DAQ from the dosing tank’s ultrasonic sensor. This research aims to develop an SVR model for predicting future liquid levels using this sensor data. Data will be collected through the TIA Portal and undergo pre-processing to ensure quality for model training. The SVR model will be trained and evaluated using metrics like Mean-squared error (MSE) and R-squared, with hyperparameter tuning employed to optimize prediction accuracy. Importantly, this research explores the potential for real-time control by integrating the trained SVR model within the digital twin. This could allow for using predicted water levels to adjust process parameters in real-time, such as regulating pump operation or valve positions, ultimately optimizing dosing tank liquid levels.

Previous research has explored the application of machine learning techniques in various industrial settings, including predictive maintenance, quality control, and process optimization. For example, developing digital twins to present the framework and workflow of the data-driven models for wind turbines [4] and using digital twin technology for production optimization in the petrochemical industry [5]. However, the integration of machine learning with digital twins to enhance sensor-based predictions and process control within industrial automation training systems remains a relatively unexplored area. This study aims to contribute to this emerging field by demonstrating the potential of SVR models in predicting liquid levels and exploring the creation of a digital twin framework for the Festo MPS PA bottling station.

The study achieved promising initial results. The SVR model integrated within the Festo MPS PA Didactic System and TIA Portal showed potential for effectively predicting liquid level. Additionally, exploring a data-driven, sensor-agnostic prediction approach and a virtual entity framework within the TIA Portal suggests promising future developments in digital twin technology for industrial automation training systems.

## 2. Related Work

Digital twins have emerged as a critical technology for enhancing industrial processes through cyber–physical integration. A comprehensive review by Tao et al. [6] provides a comparative analysis of cyber–physical systems (CPS) and digital twins, highlighting the subtle yet important differences between these two concepts. While both technologies aim to achieve seamless interaction between the physical and digital worlds, the study emphasizes the distinct capabilities and applications of digital twins in smart manufacturing. The current research aligns with this analysis by utilizing a digital twin framework, specifically tailored for predictive modeling in process automation, thus extending the scope of digital twin applications beyond traditional CPS.

Kammerer et al. [7] applied digital twins for anomaly detection in manufacturing systems, focusing on real-time data analysis to enhance predictive maintenance strategies. Although this approach effectively identified performance deviations, it did not incorporate machine learning models for continuous variable prediction, such as water levels. By integrating SVR to predict the RoC of water levels, the current study addresses this gap, particularly in mitigating error accumulation over time.

Chryssolouris et al. [8] explored digital twin technology for estimating the remaining useful life (RUL) of manufacturing equipment, leveraging physics-based simulation models. This method provided accurate predictions without disrupting operations, showcasing the potential of digital twins in predictive maintenance. However, the approach remained reliant on traditional simulations, whereas the current study adopts a data-driven methodology using radial basis function (RBF) SVR, which enhances prediction accuracy through parameter optimization.

Similarly, Putawa et al. [9] investigated the use of digital twins for visualizing and controlling energy efficiency in manufacturing environments. While effective in system control, the study focused primarily on visualization rather than predictive analytics. The current research shifts towards predictive capabilities, employing SVR within a digital twin framework to forecast water levels accurately, thereby enhancing process control and monitoring.

In summary, this research advances the application of digital twins by integrating RBF SVR for RoC prediction and optimizing SVR parameters to minimize error accumulation, thereby addressing specific challenges in process automation and control. This approach not only builds upon existing studies, but also contributes to the broader field of digital twin technology in industrial settings.

## 3. Methodology

This research employs the Festo MPS-PA Didactic Complete System’s bottling station (Esslingen, Germany) and SVR to establish a machine learning framework. The primary objective of this framework is to predict the water level in the tank, a measurement typically acquired via an installed ultrasonic sensor. An overview of the system setup is illustrated in Figure 1, providing a detailed visual representation of the configuration.

### 3.1. Festo MPS PA Bottling Station

The Festo MPS PA system comprises four stations: Filtration, Mixing, Reactor, and Bottling. These stations simulate the water filtration process, providing students with practical knowledge in Process automation and Control theory. This is achieved by controlling multiple variables, including pressure, flow rate, temperature, and water level. This study focuses on the bottling station, which houses the ultrasonic sensor. The aim is to develop a digital twin of this sensor. Further details of the experimental setup will be discussed in Section 4.

### 3.2. Human–Machine Interface

The human–machine interface (HMI), designed using Siemens’ WinCC Runtime Advanced, enables operators to control processes and monitor real-time data. It also collects operational data, including timestamps, sensor readings, and actuator values. While a DAQ device can directly obtain these values, the standard in process automation is to use the HMI. This study adheres to these industrial standards, ensuring its applicability in real-world scenarios.

### 3.3. Support Vector Regression

In this study, SVR is implemented as the machine learning technique to forecast the output of an ultrasonic sensor. The accurate prediction serves as an early warning system for potential failures, which are indicated by discrepancies between actual and predicted values [10]. SVR, a robust supervised machine learning algorithm, is specifically designed for regression tasks [11]. It generates predictions based on a set of input data points. Unlike conventional regression methods that strive to minimize the total error between predicted and actual values, SVR constructs a hyperplane in a high-dimensional space that maximizes the error margin around the most influential data points, referred to as support vectors [11]. This focus on margin minimization reduces the influence of outliers, rendering SVR particularly suitable for applications involving noisy sensor data, such as ultrasonic sensor measurements in industrial settings.

For this study, an RBF kernel was chosen for the SVR model due to its effectiveness in handling non-linear relationships that might exist between sensor readings and water level. The model development utilized the Scikit-learn library [12]. The pre-processed sensor data obtained from the HMI were divided into training and testing sets. The training set was used to train the model, while the testing set evaluated the model’s ability to generalize to unseen data. Hyperparameter tuning, employing a grid search technique, was conducted to identify the optimal combination of parameters that minimize the MSE on the validation set by using 3 nested for loops to run through all the possible combinations of hyper-parameter for the best result. The performance of the trained model was subsequently evaluated using metrics like MSE and R-squared on the testing set [13].

#### Explanation of Hyperparameters

Cost (*C*): Controls the trade-off between maximizing the margin and minimizing the training error. A higher *C* value allows less margin violation, leading to a more complex model.Gamma (γ): Influences the width of the RBF kernel function. A smaller gamma results in a wider kernel, leading to smoother decision boundaries.Epsilon (ε): Defines the width of the insensitive zone within the epsilon-SVR formulation. Points within this zone are not penalized in the loss function.

By carefully tuning these hyperparameters, the optimal SVR model for predicting water level was determined.

## 4. Experiment Setup and Implementation

### 4.1. Festo MPS PA Bottling Station

The Festo MPS PA bottling station serves as a valuable training tool in educational settings, specifically designed to introduce students to the intricacies of industrial bottling processes. While not a full-fledged industrial system itself, it effectively simulates the steps involved in filling bottles with liquid. It features the following essential components that mimic a real bottling line (Figure 2):

Reservoir tank (Area 19 cm×19 cm, Height 34 cm): This tank functions as the primary storage container for the liquid to be bottled. It has a larger capacity compared to the dosing tank and is refilled periodically to maintain a stable liquid supply.Dosing tank (Top diameter: 18 cm, Mid-diameter: 13 cm, Bottom diameter: 4 cm, Height 33 cm): This tank acts as a container between the reservoir tank and the bottling line. It has a smaller capacity than the reservoir tank and integrates with an ultrasonic sensor to precisely measure the liquid level in the tank.Pepperl+Fuchs Ultrasonic Sensor 3RG6232-3JS00-PF (Mannheim, Germany): This sensor leverages ultrasonic technology to achieve continuous, non-contact measurement of the liquid level within the dosing tank. The acquired data play a critical role in the bottling process by enabling real-time monitoring and control, directly contributing to consistent product volume in the filled bottles by preventing overflows or underfills.Johnson CM30P7-1 Pump (SPX Flow, Charlotte, NC, USA): The Festo MPS PA bottling station employs the Johnson CM30P7-1, a compact and efficient centrifugal pump, to ensure a smooth flow of liquids within the system. This pump leverages a rotating impeller to generate centrifugal force, effectively propelling the liquid through the pump housing.Gemü 524D114124DCU solenoid valve (Ingelfingen, Germany): This solenoid valve is utilized for controlled liquid dispensing during the bottling process, and facilitates precise filling through a potential adjustable flow rate mechanism and material compatibility with the processed liquid. An anti-drip design is incorporated to minimize drips or spills after dispensing, promoting cleanliness and reducing product waste.Siemens S7-300 CPU 314C-2PN/DP (Munich, Germany): The Siemens S7-300 programmable logic controller (PLC) plays a central role in automating the Festo MPS PA bottling station. It receives signals (buttons) and controls actuators (pump, valve) based on its program. The PLC makes the bottling process automated and flexible.

### 4.2. Test Cases and Experiment Settings

This section details the design and evaluation procedures employed to assess the SVR model’s performance for water prediction in the Festo MPS PA bottling station simulation (Figure 3).

#### 4.2.1. Scenarios

The test cases mimicked real-world bottling conditions by simulating various water level fluctuations that might occur during the filling process. These scenarios involved manipulating the two buttons (pump and solenoid valve) and the DAQ button to represent diverse operational conditions.

Filling cycle variations: Test cases included different button press combinations and duration for the pump and valve buttons. This simulated variations in water transfer volumes between tanks and diverse filling operations.DAQ: Users could define the time interval between data points logged in the CSV file via the DAQ button. This allowed for customization based on the desired data granularity for model training.

#### 4.2.2. Settings

The experiment setup involved a two-pronged approach:
TIA Portal Program: The operation logic and communication between components were programmed within the TIA Portal. This program included functions for:–Button control: Responding to user interaction with the pump and valve buttons, triggering virtual button presses within the WinCC environment.–Sensor communication: Establishing communication with the ultrasonic sensor to retrieve real-time water level readings.–Data transfer: Potentially transferring the collected water level data to WinCC for further processing or visualization.WinCC Advanced Runtime Environment: WinCC Advanced RT provided the platform for controlling the experiment, acquiring data, and triggering data logging scripts. The functionalities within WinCC were achieved through the Visual basic (VB) scripts.

#### 4.2.3. SCADA Design and Data Logging Using VB Script

The experiment leveraged a supervisory control and data acquisition (SCADA) design approach within WinCC Advanced Runtime to control the simulation and acquire data for model development. VB scripts played a vital role in this SCADA design, acting as the interface between the HMI and the physical components.

SCADA design elements:–HMI control: The VB scripts provided functionalities displayed on the HMI, such as buttons for pump and valve activation, and a DAQ button.–DAQ: The scripts interacted with the ultrasonic sensor through WinCC tags, retrieving water level readings at a defined interval.–Data logging: Upon pressing the DAQ button, the script triggered data logging into a CSV file.VB script functionalities:–Simulation and Control: The scripts facilitated precise control over the simulated filling cycles by automatically triggering virtual button presses based on the defined scenarios.–DAQ: The scripts interacted with the ultrasonic sensor through WinCC tags to retrieve real-time water levels at specified intervals throughout the simulated filling cycles.–Data logging: When pressed, the DAQ button will activate the scripts to retrieve the collected data and write them along with timestamps to a user-defined location in a CSV format. This enabled easy import into data analysis tools for model training and evaluation.

### 4.3. Format of Data

The VB scripts generated a CSV file for model training and evaluation. This file offers a well-defined format for efficient processing by the SVR model. Each column of data is separated by a semicolon (;) delimiter. The data format includes:Timestamps: Captures the time of each data point (e.g., YYYY-MM-DD HH:MM:SS) for time-based analysis.Water Level Reading: Represents the real-time water level measurement (milliliters)—the target variable for prediction.Input Features of the VB screen in WinCC:–Pump: Toggle the ON and OFF state of the 4M1 Pump;–Valve: Toggle the ON and OFF state of the 4M2 Valve;–Start data logging: Start the VB script that logs the data on the WinCC server side;–Other input data are controlled by directly modifying the values in TIA Portal’s HMI tag table.

By adhering to this structured format with semicolon delimiters, the CSV file provides a well-organized dataset suitable for machine learning model training and evaluation.

## 5. Model Development

### 5.1. Lagrange Duality

Lagrange duality is a fundamental concept in convex optimization and plays an important role in an SVR algorithm [14]. An optimization often has a standard form as follows:(1)$$\begin{array}{cc} \mathrm{minimize}\; & f_{0}\left(x\right)\\ \mathrm{subject}\;\mathrm{to}\; & \left\{\begin{array}{c} f_{i}\left(x\right)\leq 0,i=0,\ldots m\\ h_{i}\left(x\right)=0,i=0,\ldots p \end{array}\right. \end{array}$$

A Lagrangian approach is to optimize Equation (Equation 2) instead:(2)$$\underset{x}{\inf}P\left(x\right)=\underset{x}{\inf}\underset{\lambda \geq 0}{\sup}L\left(x,\lambda \right)=\underset{x}{\inf}\underset{\lambda \geq 0}{\sup}\left(f\left(x\right)+\sum_{k}\lambda_{k}f_{i}\left(x\right)\right)$$
where $L\left(x,\lambda \right)=f\left(x\right)+\sum_{k}\lambda_{k}f_{i}\left(x\right)$, $P\left(x\right)=\sup_{\lambda \geq 0}L\left(x,\lambda \right)$.

Equation (Equation 2) is referred to as the Primal Lagrangian equation. The second alternative to the Primal function is the Dual Form equation:(3)$$\underset{\lambda \geq 0}{\sup}D\left(\lambda \right)=\underset{\lambda \geq 0}{\sup}\underset{x}{\inf}L\left(x,\lambda \right)\leq \underset{x}{\inf}\underset{\lambda \geq 0}{\sup}L\left(x,\lambda \right)=\underset{x}{\inf}P\left(x\right)$$
where $D\left(\lambda \right)=\inf_{x}L\left(x,\lambda \right)$.

### 5.2. Support Vector Machine with Linear Kernel

Given a linear kernel, a regression line will have a general linear function in the form of Equation (Equation 4).(4)$$\langle w,X_{i}\rangle +b-y_{i}=0$$

The objective of SVR is to determine the vector w such that the resulting linear function ensures most of the data points $\left(X_{i},y\right)$ remain within a specified ε tolerance. Figure 4 shows the general presentation of the kernel in the case that w and $X_{i}$ only have one dimension, but in application, both of these vectors can have much higher dimensions that cannot be represented in a Cartesian coordinate.

The aim of SVR now becomes Equation (Equation 5)(5)$$\begin{array}{cc} \mathrm{minimize}\; & \frac{1}{2}{∥w∥}^{2}\\ \mathrm{subject}\;\mathrm{to}\; & \left\{\begin{array}{c} y_{i}-\langle w,X_{i}\rangle -b\leq \varepsilon \\ \langle w,X_{i}\rangle +b-y_{i}\leq \varepsilon \end{array}\right. \end{array}$$

Equation (Equation 5) assumes that the convex optimization for a linear kernel function f(X)=〈w,X〉+b with an error of ε is feasible. Sometimes, this may not be the case and some amount of errors needs to be taken into account. Hence, the slack variables $\xi_{i}$ and $\xi_{i}^{*}$ need to be used [11].(6)$$\begin{array}{cc} \mathrm{minimize}\; & \frac{1}{2}{∥w∥}^{2}+C\sum_{i=1}^{l}\left(\xi_{i}+\xi_{i}^{*}\right)\\ \mathrm{subject}\;\mathrm{to}\; & \left\{\begin{array}{c} y_{i}-\langle w,X_{i}\rangle -b\leq \varepsilon +\xi_{i}\\ \langle w,X_{i}\rangle +b-y_{i}\leq \varepsilon +\xi_{i}^{*}\\ \xi_{i},\xi_{i}^{*}\geq 0 \end{array}\right. \end{array}$$

The parameter *C* determines the trade-off between the flatness of *f* and determines how much amount of deviation from the target error ε is tolerated [11].

### 5.3. Gaussian Kernel

RBF kernel regression, or sometimes called Gaussian kernel regression, is a popular algorithm that is widely used in SVM tasks or, in this case, SVR [15].

The kernel function for two sets of parameters is defined in Equation (Equation 7)(7)$$K\left(X,X_{i}\right)=\exp\left(-\gamma ∥X-X_{i}{∥}^{2}\right)$$
where $\gamma =\frac{1}{2\sigma^{2}}$, and σ is the standard deviation parameter of the Gaussian curve, which also determines the width of the Gaussian kernel [15].

The weight $w_{i}$ used in support vector regression is determined based on the similarity between the kernel *K* of data *i* and the rest of the data points.(8)$$w_{i}=\frac{K(X,X_{i})}{\sum_{j=1}^{N}K\left(X,X_{j}\right)}.$$

Implementing the kernel into an SVM system, Equation (Equation 9) is obtained(9)$$\begin{array}{cc} \mathrm{minimize}\; & \frac{1}{2}{∥w∥}^{2}+C\sum_{i=1}^{l}\left(\xi_{i}+\xi_{i}^{*}\right),\\ \mathrm{subject}\;\mathrm{to}\; & \left\{\begin{array}{c} y_{i}-\sum_{j=1}^{N}\alpha_{j}K\left(X_{i},X_{j}\right)\leq \varepsilon +\xi_{i},\\ \sum_{j=1}^{N}\alpha_{j}K\left(X_{i},X_{j}\right)-y_{i}\leq \varepsilon +\xi_{i}^{*},\\ \xi_{i},\xi_{i}^{*}\geq 0. \end{array}\right. \end{array}$$

### 5.4. Hyperparameters Explanation

The *C* parameter is a scalar that controls the penalty for the values that deviate from the predicted regression curve. The higher the value *C* is, the more noticeable the error is [12]. In contrast, a smaller value of *C* allows for a larger margin and more tolerance for errors, which can lead to a simpler model with potentially higher bias but lower variance.

The γ value controls the width of the Gaussian kernel function used for the regression process. This parameter defines the extent to which a single training example influences the model’s decision boundary [12]. A lower γ results in a wider influence, making each training example affect a larger area of feature spaces.

ε is the margin of tolerance around the predicted curve. If a data point is positioned within this margin, the point will have 0 error penalty [12]. This margin will act as an error threshold. Similar to the *C* parameter, a larger ε will result in a higher bias and lower variance predicted curve.

### 5.5. Model Explanation

Based on the experimental setup illustrated in Figure 3, the network’s input layer comprises six variables, as detailed in Table 1.

Upon initial observation using Microsoft Excel 365 (Version 2312, Microsoft) [16] graphing tool, utilizing the RoC of the water level as training data rather than the water level itself appears to be more effective for the labeling process. This is due to the direct correlation observed between the valve state (0 and 1), Button and Pump values, and the RoC of the water level.

The discrete RoC of the parameter is calculated as follows:(10)$$\frac{\Delta y}{\Delta t}=\frac{y_{i}-y_{i-1}}{1s}=y_{i}-y_{i-1}.$$

The Time variable is relabeled manually with respect to Button1 (Figure 5) to make the learning process more effective. This is due to Button1 having the most influence over the predicted data. Each time Button1 goes from 0 to 1 and to 0 again, the Time variable is reset to 0, creating an ON and OFF cycle, which is equal to splitting one dataset into many smaller datasets for better pattern recognition.

## 6. Model Evaluation and Chosen Hyperparameters

In this experiment, the true water level measured using the ultrasonic sensor will be assigned to Y, and the water level predicted by the algorithm will be stored to pred_Y. Similarly, the RoC of the water level derived from Y is stored in dY, and the predicted RoC of the water level (which will be used to calculate pred_Y) is assigned to pred_dY. The experiment was performed five times and produced no significant difference (less than 0.01%) for each set of hyperparameter *C*, ε, and γ.

When the predicted curve undergoes the integration process later on in the network, any slight deviation between dY and pred_dY will stack up overtime. Evidently, even though the error is not observed to be a problem in Figure 6, the difference between Y and pred_dY keeps increasing continuously in Figure 7.

This makes it more effective to use a higher *C* and lower ε value for the training process. Lowering ε will decrease the tolerance margin for the error value, and increasing the *C* will increase the penalty of each predicted value that deviates further from the ε-margin. This will force pred_dY to be much closer to the value of dY. In turn, this will also increase the variance of pred_dY, as discussed in Section 5.4. This should not be a problem in this particular use case because the integration process will smooth out most of the variance as long as dY does not change rapidly from positive to negative values.

The chosen hyperparameters for the problem are: C=320, γ=0.72, ε=1.3.

Upon comparison between Figure 6 and Figure 8, it becomes clear that the adjusted *C*, γ, and ε values have greatly decreased the bias between the dataset and the predicted curve. Particularly, values from t=2→5,12→25,27→31s have a larger influence over pred_dY. This helps largely reduce the deviation in the integration process, as shown in Figure 9.

pred_dY may have high variation, but a large or small positive value still results in an increase in dY. Similarly, a large or small value of negative pred_dY will result in a decrease in dY. This characteristic can be demonstrated by taking the average error of Figure 8b and b. Even though the SVR model showed an average 18.38% error rate for predicting RoC of the water level, it only showed an average 6.99% error rate for predicting the actual water level after integration.

## 7. Conclusions

In conclusion, the evaluation of the sensor output prediction model, the optimization of hyperparameters, and the utilization of derivation and integration have provided an increase in accuracy for the sensor behavior predictions.

Subsequent to this experiment, it is advised to integrate more parameters into the system. Specifically, the application of simulation coupling could optimize computational resources and reduce inference time [17]. This addition would introduce another layer to the network and alleviate the drifting error that occasionally cannot be eradicated solely by predicting the derivative [18]. Furthermore, the incorporation of sensor fusion algorithms, such as the Kalman filter, could significantly enhance the model’s precision by consolidating data from diverse sources [19]. This strategy offers a more resilient and precise prediction model for future applications.

Lastly, to visualize and analyze the system holistically, a digital twin representation can be constructed using CAD software like Siemens’ NX for the ultrasonic sensor and a three-dimensional visualization engine like Tecnomatix Plant Simulation for the entire Festo MPS PA system. This approach could serve as a foundation for digitizing various manufacturing processes [20].

## Figures and Tables

**Figure 1 sensors-25-00678-f001:**
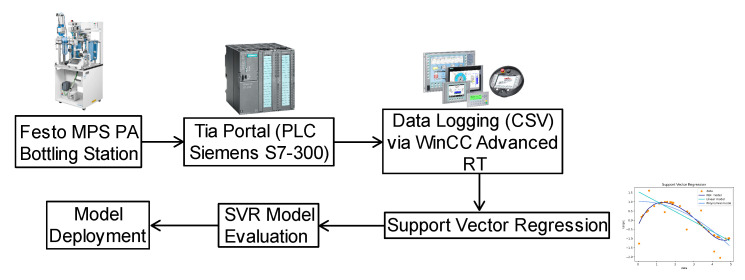
Overview of the proposed modeling method.

**Figure 2 sensors-25-00678-f002:**
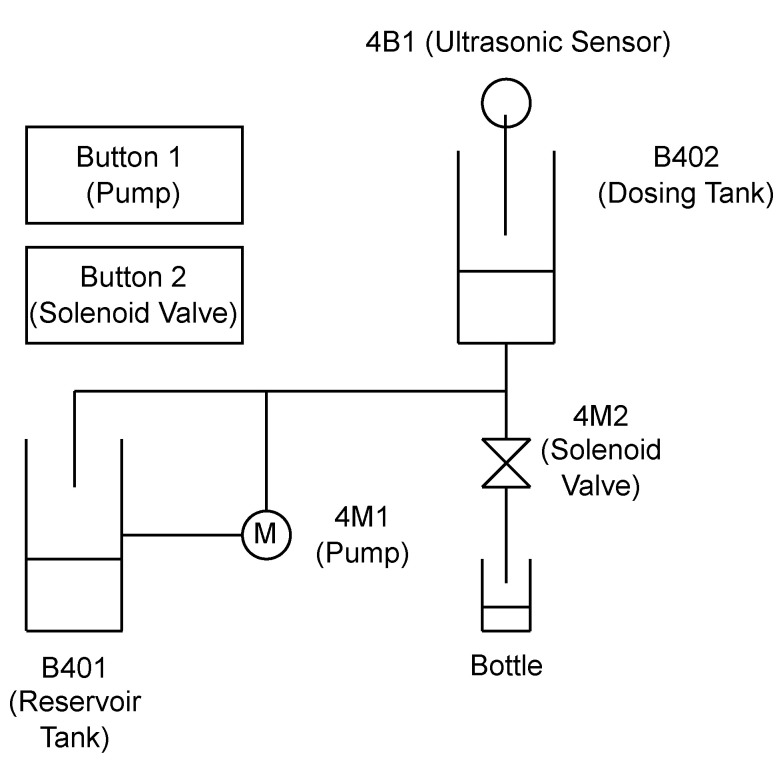
Festo MPS PA bottling station.

**Figure 3 sensors-25-00678-f003:**
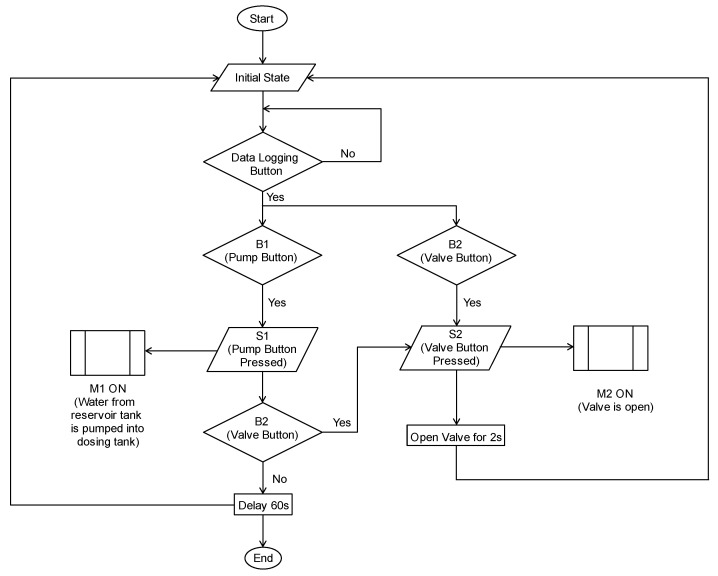
Experiment flowchart.

**Figure 4 sensors-25-00678-f004:**
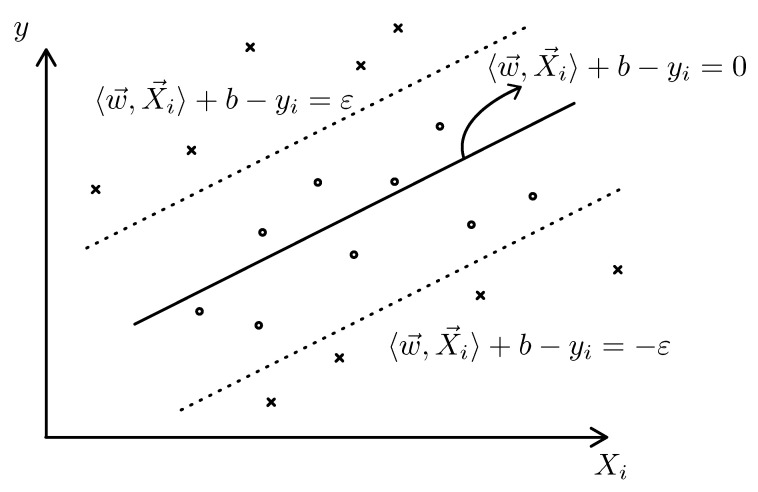
Regression using a linear kernel: The data points marked with ‘x’ fall outside the acceptable tolerance, while the data points marked with ‘o’ lie within the acceptable error range.

**Figure 5 sensors-25-00678-f005:**
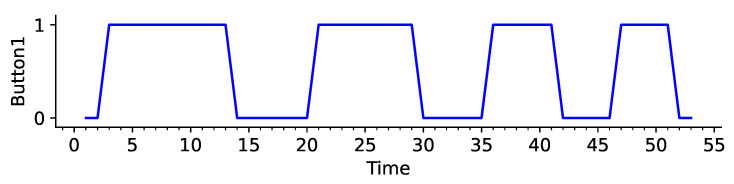
Value of Button1 with respect to Time in seconds.

**Figure 6 sensors-25-00678-f006:**
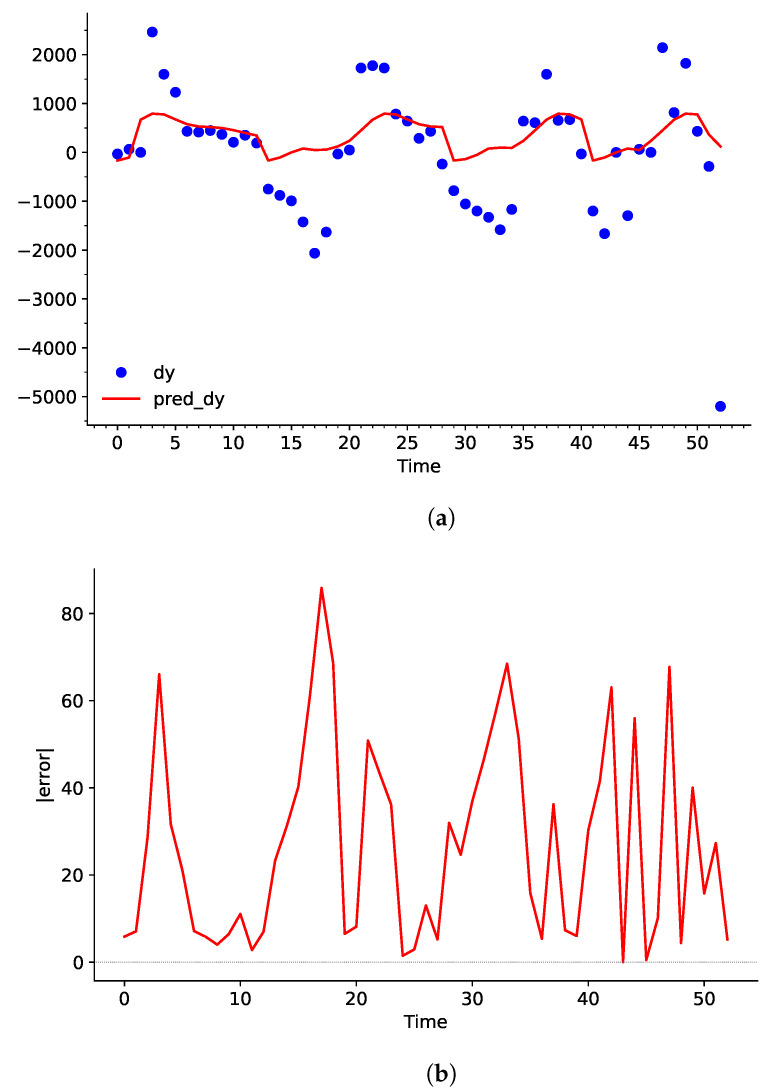
(**a**) Plot of dY and the predicted curve pred_dY using conventional hyperparameters, and (**b**) the error in percentage with respect to Time in seconds.

**Figure 7 sensors-25-00678-f007:**
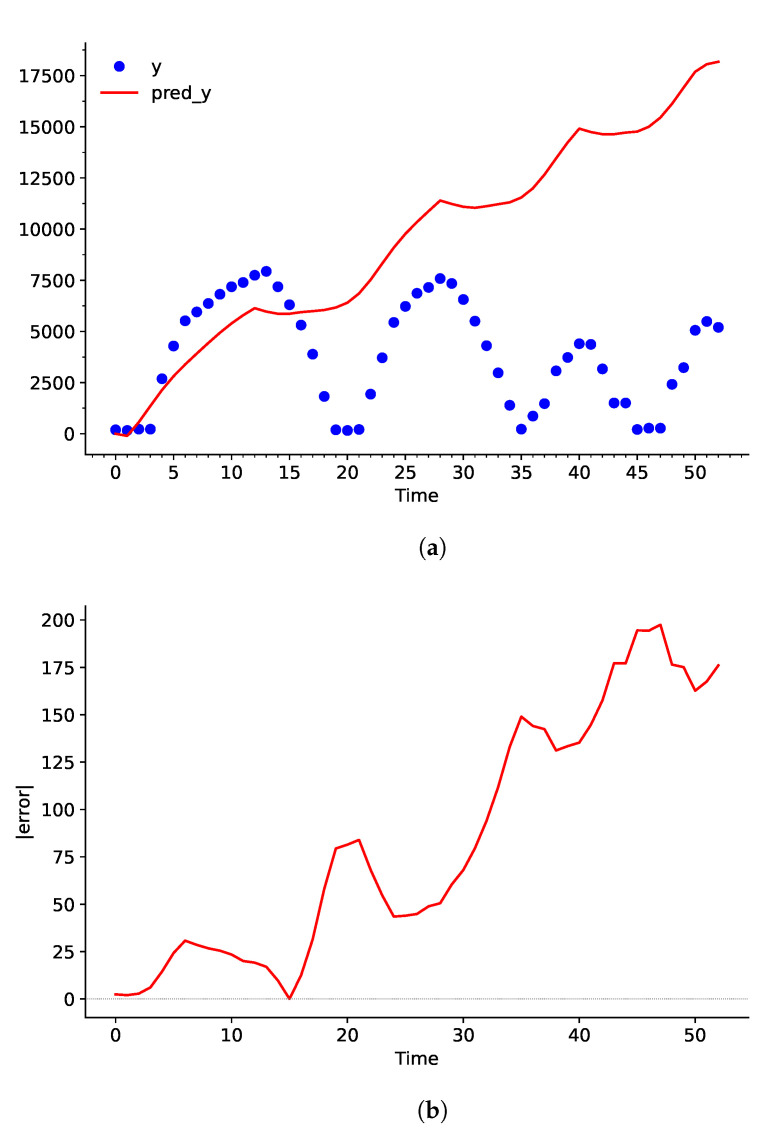
(**a**) Plot of Y and the predicted curve pred_Y using conventional hyperparameters, and (**b**) the error in percentage with respect to Time in seconds.

**Figure 8 sensors-25-00678-f008:**
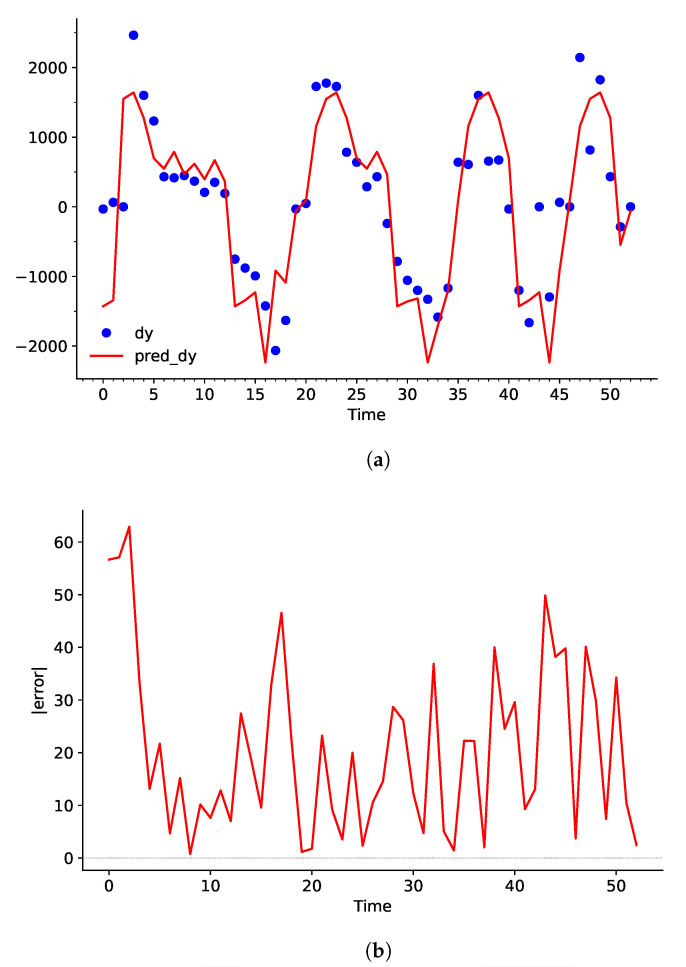
(**a**) Plot of dY and the predicted curve pred_dY using adjusted hyperparameters, and (**b**) the error in percentage with respect to Time in seconds.

**Figure 9 sensors-25-00678-f009:**
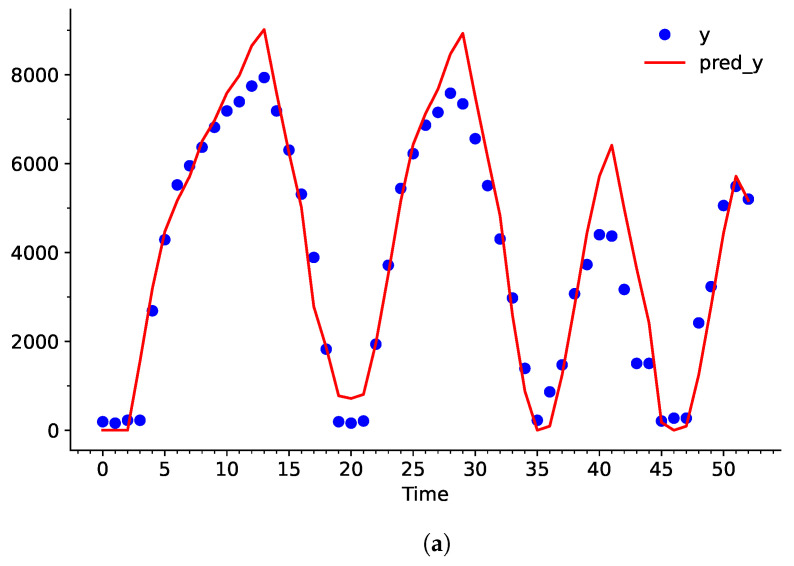
(**a**) Plot of Y and the predicted curve pred_Y using adjusted hyperparameters and (**b**) the Error in percentage with respect to Time in seconds.

**Table 1 sensors-25-00678-t001:** Input variables of the model.

Variable	Data Type	Range
Time	Integer, in seconds	0 to +∞
Button 1	Boolean	ON and OFF
Button 2	Boolean	ON and OFF
Valve State	Boolean	ON and OFF
Pump State	Boolean	ON and OFF
Pump Analog	Integer	0 to 27,648

## Data Availability

Dataset available on request from the authors.

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
