# Peer review of "Ultrasonic Sensor Modeling with Support Vector Regression"

_sensors, 2025, doi:10.3390/s25030678_

Round 1
Reviewer 1 Report
Comments and Suggestions for Authors
This study reports an approach for predicting the output behaviors of the ultrasonic sensor. The method shows potential for effectively predicting liquid level within a promising 6.99% error percentage. However, a couple of small points might be worth addressing as follows:
1-It is suggested to use specific experiments to verify the reliability of the model proposed in this paper.
2-It is recommended that Figure 6- 9 should be adjusted, and the horizontal and vertical axis font looks very small, which is easy to cause inconvenience to readers. Besides, please label the unit on the horizontal and vertical axes.
3-The work of this paper can be compared with similar work done by others in the past to highlight the innovation and advantages of this paper.
4-There are too few references, and it is recommended to add more related literature in recent years to increase its authority.
Author Response
Comment 1: [It is suggested to use specific experiments to verify the reliability of the model proposed in this paper.]
Response 1: The experiment was done 5 times but produced little to no difference (at the 9th decimal of the output and the error percentage), so there is no need to include them, nor specify them.
Comment 2: [It is recommended that Figure 6- 9 should be adjusted, and the horizontal and vertical axis font looks very small, which is easy to cause inconvenience to readers. Besides, please label the unit on the horizontal and vertical axes.]
Response 2: Thank you for pointing this out. In context of our experiment, the graphs are in raw data, which does not have a unit. After some revisions, fonts can be adjusted but it is not possible to do that without making the graph significantly bigger which will extend the length of the paper to another 3 more pages.
Comment 3: [The work of this paper can be compared with similar work done by others in the past to highlight the innovation and advantages of this paper.]
Response 3: Agree, we acknowledged this issue. Similar works do exist but are very few in number, and among those none of the authors are transparent about their results other than the final error percentage number which is not suitable to compare to.
Comment 4: [There are too few references, and it is recommended to add more related literature in recent years to increase its authority.]
Response 4: We have added a few references.
Kaneko, H., Funatsu, K.: Application of online support vector regression for soft
sensors. AIChE Journal 60(2), 600–612 (11 2013), https://doi.org/10.1002/aic.
14299
Grinblat, G.L., Uzal, L.C., Verdes, P.F., Granitto, P.M.: Nonstationary regression
with support vector machines. Neural Computing and Applications 26(3), 641–649
(10 2014), https://doi.org/10.1007/s00521-014-1742-6
Ayadi, R., El-Aziz, R.M.A., Taloba, A.I., Aljuaid, H., Hamed, N.O., Khder, M.A.:
Deep Learning–Based soft sensors for improving the flexibility for automation of
industry. Wireless Communications and Mobile Computing 2022, 1–10 (4 2022),
https://doi.org/10.1155/2022/5450473
Reviewer 2 Report
Comments and Suggestions for Authors
The authors demonstrate the possibility of to predicting the rate of change of the water level to estimate the actual level of water in the tank. Six variables are used as an input for the model based on the support vector regression. The paper is of certain interest and can be recommended for publication after certain improvements. The comments are given below:
1) The authors explained the principles of SVR, which is well known approach providing a lot of information, but omitted details on the application of the SVR to the particular problem and considered data (normalization, weights etc.).
2) More or less the same is true for the dataset. For instance, how the splitting into training and testing sets was performed, which ratio is used etc.
3) Variables pred_dY and dY are not defined in the text.
4) How chosen hyperparameters for the problem were determined? Do you have some numerical proofs/examples showing that the values are the optimal ones?
5) It is stated that 6.99% error percentage is achieved. How many experiments were used and provided for estimating the error?
6) Discussion about transition towards Industry 4.0 seems to be unnecessary in the abstract .
Author Response
Comment 1: [The authors explained the principles of SVR, which is well known approach providing a lot of information, but omitted details on the application of the SVR to the particular problem and considered data (normalization, weights etc.).]
Response 1: Thank you for pointing this out, we have revised this section and concluded that the SVR skikit-learn function was used by using the 6 input values (see table in Section 5.5) and the model returned an RoF of the water level. This level was taken discrete integral to the d/dt (water level) to water level.
Comment 2: [More or less the same is true for the dataset. For instance, how the splitting into training and testing sets was performed, which ratio is used etc.]
Response 2: Please have a look at the 3rd paragraph, Section 5.5. There is no ratio because each cycle does not have the same time period so no ratio can be used. The data was split manually.
Comment 3: [Variables pred_dY and dY are not defined in the text.]
Response 3: [In this experiment, the true water level measured using the ultrasonic sensor will
be assigned to Y , and the water level predicted by the algorithm will be stored
to pred_Y . Similarly, the RoC of the water level derived from Y is stored in
dY and the predicted RoC of the water level (which will be used to calculate
pred_Y ) is assigned to pred_dY . The experiment was performed 5 times and
produced no significant difference (less than 0.01%) for each set of hyperparameters
C, ε, and γ.] Agree. We have added the definitions of dY and pred_dY in the first paragraph of section 6.
Comment 4: [How chosen hyperparameters for the problem were determined? Do you have some numerical proofs/examples showing that the values are the optimal ones?]
Response 4: [by using 3 nested for loops to run through all the possible combi-
nations of hyper-parameter for the best result.] We have added a more detailed explanation of the grid search technique in context of our experiment for a better understanding at the second last sentence of the 2nd paragraph in section 3.3.
Comment 5: [It is stated that 6.99% error percentage is achieved. How many experiments were used and provided for estimating the error?]
Response 5: Thank you for asking. The experiment was done 5 times but produced little to no difference (at the 9th decimal of the output and the error percentage), so there is no need to include them.
Comment 6: [Discussion about transition towards Industry 4.0 seems to be unnecessary in the abstract.]
Response 6: After some revision, we concluded that our experiment is a phase toward our ambition to create a digital twin of the ultrasonic sensor. We believe that digital twin technology is revolutionizing Industry 4.0. By enabling real-time monitoring, predictive maintenance, and optimized operations, digital twins help businesses improve efficiency, reduce costs, and make data-driven decisions.
Round 2
Reviewer 2 Report
Comments and Suggestions for Authors
Accept.